# Rearrangement of the Cellulose-Enriched Cell Wall in Flax Phloem Fibers over the Course of the Gravitropic Reaction

**DOI:** 10.3390/ijms21155322

**Published:** 2020-07-27

**Authors:** Nadezda Ibragimova, Natalia Mokshina, Marina Ageeva, Oleg Gurjanov, Polina Mikshina

**Affiliations:** Kazan Institute of Biochemistry and Biophysics, FRC Kazan Scientific Center of RAS, Lobachevsky Str., 2/31, 420111 Kazan, Russia; nibra@yandex.ru (N.I.); ne.mokshina@gmail.com (N.M.); mageeva58@mail.ru (M.A.); gurjanov@kibb.knc.ru (O.G.)

**Keywords:** *Linum usitatissimum* L., gravistimulation, fiber cell wall, rhamnogalacturonan I, methylated homogalacturonan, galactosylated xyloglucan, cellulose, crystallinity index, mobility of cellulose, solid-state NMR

## Abstract

The plant cell wall is a complex structure consisting of a polysaccharide network. The rearrangements of the cell wall during the various physiological reactions of plants, however, are still not fully characterized. Profound changes in cell wall organization are detected by microscopy in the phloem fibers of flax (*Linum usitatissimum*) during the restoration of the vertical position of the inclined stems. To characterize the underlying biochemical and structural changes in the major cell wall polysaccharides, we compared the fiber cell walls of non-inclined and gravistimulated plants by focusing mainly on differences in non-cellulosic polysaccharides and the fine cellulose structure. Biochemical analysis revealed a slight increase in the content of pectins in the fiber cell walls of gravistimulated plants as well as an increase in accessibility for labeling non-cellulosic polysaccharides. The presence of galactosylated xyloglucan in the gelatinous cell wall layer of flax fibers was demonstrated, and its labeling was more pronounced in the gravistimulated plants. Using solid state NMR, an increase in the crystallinity of the cellulose in gravistimulated plants, along with a decrease in cellulose mobility, was demonstrated. Thus, gravistimulation may affect the rearrangement of the cell wall, which can enable restoration in a vertical position of the plant stem.

## 1. Introduction

It is well-known that higher plants, as a rule, are unable to move their whole bodies. Only movements of their organ parts are possible. These movements are determined by the sensitivity of the plant to external stimuli and are provided by different types of tropisms [1,2]. In the case of gravitropism, the external stimulus that causes a motor reaction is gravity [1,2,3,4]. The force of gravity provides a constant background of mechanical stimulation for the plant. Gravitropism is traditionally divided into three major phases: sensing, transmission, and response [5]. Despite the long-term studies of this phenomenon, there are many more questions than answers regarding the mechanisms implemented in the course of all three phases including the motor reaction [4,6].

Herbaceous plants restore their position in space mainly due to the asymmetric growth of the inclined stem. The differential growth response is induced by the redistribution of plant hormones [1,4]. The concept of cooperative hormone action during graviresponse (not only auxin) suggests the complex interactions of all phytohormones and their possible synergistic effects and involvement in the gravitropic bending process [7]. Previously, we showed that the implementation of the gravitropic reaction in herbaceous plants could be caused by another mechanism that does not include cell growth via elongation. Even after removal of the upper part of a flax plant that underwent elongation, the plant successfully carried out a gravitropic reaction by restoring its position in space [8]. One of the most well-known examples of the gravity response is the formation of reaction wood (tension wood) on the upper side of gravistimulated stems in angiosperms where this response generates a tensile force that pulls the stem upward [9,10,11,12]. Tension wood is often characterized by the formation of fibers with smaller diameters containing a gelatinous layer after the second layer (S2) of the secondary cell walls where cellulose microfibrils are aligned in a vertical orientation [13] with a lower lignin content [14].

Previously, using various approaches of microscopy, we analyzed the flax stem curvature region and found significant modifications in the primary phloem fibers in the cell wall of which the gelatinous layer is formed constitutively. Within the formed stem curvature, the pulling side toward which the plant restored its position could be distinguished from the opposite side. Although the reaction was accompanied by the appearance of a gelatinous layer in the xylem fibers [8], drastic effects were observed in the phloem fibers located on the pulling side of the stem. The lumen diameter increased while the cell wall thickness decreased. Additionally, the cell portions were widened with the formation of callose-containing “bottlenecks” between them, which leads to a “sausage-like” shape of the cell [8]. Transcriptomic analysis of the isolated fibers of gravistimulated plants revealed the up-regulation of the genes encoding the cell wall proteins (fasciclin-like arabinogalactan proteins, expansins, extensins, inhibitors of pectin methylesterases, etc.), some isoforms of cellulose synthases, and their cofactors [15,16]. Among the genes involved in the metabolism of plant hormones and in signalling, genes for enzymes of different steps of gibberellin biosynthesis belonged to the most upregulated [15]. Flax phloem fibers form a thickened tertiary cell wall (gelatinous cell wall, G-layer) characterized by a high content of cellulose with microfibrils locked in an axial direction with a lack of lignin and xylan and the presence of pectins, xyloglucan, mannans, and arabinogalactan [17,18,19]. The tertiary cell wall is formed in the fibers of plant crops (hemp, ramie) and tension wood [19,20]. The specific supramolecular arrangement of cellulose microfibrils and non-cellulosic polysaccharides, whose content is 5–15% [17,19], their interactions, and their post-synthetic modifications may provide flax phloem fibers in which the contractile properties that allow them to implement their mechanical functions [19] is similar to xylem fibers in tension wood [13,21]. Although the role of pectic rhamnogalacturonan I in this system has been well-characterised [19,22], the role of other non-cellulosic polysaccharides, the way of their interactions with each other, and cellulose are not yet established. Moreover, there are no studies describing changes in the fine structure (crystallinity, accessible/inaccessible fibril surface, I_α_/I_β_) and mobility of the cellulose molecules caused by gravistimulation. We proposed flax phloem fibers as a model to study the mechanism of the motor reaction of higher herbaceous plants in vivo [8]. This model could be useful for studying the fundamental physiological processes induced by an external stimulus (perception of the mechanical stimulus, the activation of gravisensitive proteins and ion transporters, etc.) as well as for a deeper understanding of the cell wall’s structure and its rearrangement during the graviresponse since, due to the specific properties of the fiber cell wall (among other properties), flax plants are able to restore the position of their whole stem after inclination.

In the current study, the content and arrangement of the matrix polysaccharides and structural peculiarities of cellulose in the cell wall of phloem fibers taken from the pulling side of the stems of gravistimulated plants were compared with the fiber cell walls of non-inclined plants. Since we used the lower part of the flax stem containing the fibers at an advanced stage of development, we were able to obtain pure isolated phloem fibers with thickened tertiary cell walls for biochemical and NMR analysis. Using different methods (microscopy, solid state NMR, and chromatography), we characterised changes in the cell wall structure and organization that could be useful for further investigations of the role of post-synthetic modifications of the plant cell wall and role of minor non-cellulosic polysaccharides in the plant’s biomechanical properties.

## 2. Results

To identify the peculiarities of polysaccharides inherent to gravitropic reactions, the total monosaccharide composition, distribution of various types of polysaccharides, and crystalline and amorphous cellulose in the fiber cell walls of gravistimulated (pulling part) and non-gravistimulated plants were compared. The scheme for sample collection and analysis is presented in Figure 1.

### 2.1. Total Monosaccharide Composition of the Control and Gravistimulated Flax Plants

To estimate the total monosaccharide composition, the hydrolysates of flax phloem fiber obtained after trifluoroacetic acid (TFA) treatment and subsequent treatment with sulfuric acid were analysed. Glucose was the main monomer in both the TFA and sulfuric acid hydrolysates of all samples (Figure 2). The largest portion of glucose was released after the complete hydrolysis of the TFA pre-treated residue with sulfuric acid (Figure 2A,C). This glucose is attributed mainly to disordered cellulose. The presence of xylose in the composition of sulfuric acid hydrolysates suggests that some portion of glucose could belong to the xyloglucan fragments linked with cellulose. The predominance of glucose in the TFA hydrolysates suggests that some portion of amorphous and/or less-ordered cellulose was hydrolysed by this acid (Figure 2A). The proportion of cellulose released from the isolated fibers of the control and gravistimulated flax plants by acid hydrolysis was comparable and amounted to about 90% of the entire polysaccharide content of the cell wall.

Much more pronounced differences between the control and inclined plants were revealed in the content of non-cellulosic polysaccharides. The largest portion of non-cellulosic polysaccharides was released by TFA (Figure 2A). This portion included galactose, galacturonic acid, mannose, and xylose. The most noticeable changes in the monosaccharide proportion were observed in the TFA hydrolysates and consisted of an increase in galactose and galacturonic acid for the pulling part of the gravistimulated flax plants (Figure 2B). The total yield of non-cellulosic polysaccharides (galactans, homogalacturonan, xyloglucan, and mannan) was approximately 1.4 times higher in the cell walls of the fibers from the pulling side of the inclined stem when compared to the control plants (Figure 2A). To identify the participation of these polysaccharides in a possible modification of the cell wall architecture over the course of the gravitropic reaction, their distribution in the cell wall was analyzed using immunocytochemistry.

### 2.2. Immunolocalization of Non-Cellulosic Polysaccharides in the Cell Wall of the Control and Gravistimulated Flax Plants

#### 2.2.1. Galactan

For galactan labeling, the monoclonal antibody LM5 [23], specific to the linear tetrasaccharide of β-(1,4)-d-galactose, was used. Cell walls of both the control and gravistimulated plants were labeled with the LM5 antibody (Figure 3). Generally, the cell wall in the widened fibers from the pulling side of gravistimulated plants was always well-labeled by LM5 throughout most of its thickness, while, in the fiber cell wall of the control plants, the inner layer of the tertiary cell wall was labeled (Figure 3).

#### 2.2.2. Homogalacturonan

In the current work, we compared immunolabeling with JIM5 and JIM7 monoclonal antibodies, which recognize low and high methyl-esterified homogalacturonans (HGs), respectively [24]. The distribution pattern of both antibodies in the control plants and pulling parts of the stems was similar. The labeling of low-methylated HG was weaker when compared to highly-methylated HG and was more pronounced in the cell corners. Furthermore, epitopes for highly-methylated HG were revealed in the primary cell wall and middle lamellae of both samples (Figure 4I–P).

A slight increase in the intensity of labeling with JIM7 antibodies specific to high methyl-esterified homogalacturonan was observed in the cell walls of fibers from the pulling side on the fourth day after inclination. For the gravistimulated plants, a part of the label was also found in the outer layer of the thickened gelatinous cell wall of the fibers in the pulling side (Figure 4O,P). There were no significant differences in the intensity of cell wall labeling with JIM5 between samples (Figure 4Q). The slight and fragmentary labeling by JIM5 in the G-layer of the cell wall of the fibers in the pulling side of the stem was observed only on longitudinal sections and under significant magnification (Figure 4G). There were also no clear differences in the distribution of low-methylated HG in the control and gravistimulated plants detected by electron microscopy (Figure 4D,H). Very weak labeling was detected through the G-layer. Stronger labeling in the middle lamellae, primary cell wall, and (partially) the G-layer was detected using JIM7 (Figure 4L,P).

#### 2.2.3. Xyloglucan

Two monoclonal antibodies, LM15 and LM25, specific for xyloglucan fragments, were used in this work. LM15 recognises the XXXG motif of xyloglucan [26], whereas LM25 binds to the galactose-containing (XXLG and XLLG) oligosaccharides of this polymer [27]. To exclude the possible masking of xyloglucan epitopes *in muro* by galactans, labeling with LM15 and LM25 antibodies was carried out for a portion of the cross-sections after their preliminary treatment with pectin lyase or β-(1,4)-galactanase.

The LM15 label was not detected in the tertiary cell wall either in the control or in the gravistimulated plants even after the removal of pectin or galactans accessible to the enzymes (data not shown). The use of LM25 allowed us to detect xyloglucan in the G-layer of the fiber cell wall in the pulling side of the stem of the gravistimulated plants even without pretreatment of the cross-sections with galactanase (Figure 5B), while on the cross-sections of the control plants LM25 labeled a very thin layer of fiber cells and cells of xylem that can likely be assigned to the layer of the primary cell wall, which is usually enriched in xyloglucan (Figure 5A). The removal of galactans moreover led to the binding of LM25 to the gelatinous cell wall in the control plant (Figure 5C,E). The intensity of labeling the fibers in the pulling side after the galactanase treatment of the cross-sections was increased when compared to sections without treatment and was slightly more pronounced than that in the control plants treated with the enzyme (Figure 5C–E).

### 2.3. Crystalline and Amorphous Cellulose in the Fiber Cell Wall After Gravistimualtion

Analysis of the fine structure of cellulose in the gravistimulated and non-gravistimulated flax plants was carried out using solid state NMR and immunocytochemistry.

#### 2.3.1. Immunocytochemistry

To detect the crystalline and amorphous cellulose in the phloem fiber cell walls of the analysed plants, two different types of carbohydrate binding modules (CBMs), CBM3a and CBM28 [28], were used, respectively. Both forms of cellulose were revealed on the longitudinal sections of both samples (Figure 6). There were no significant differences between the characteristics of the cell wall labeling of the gravistimulated and control plants using CBM28 (Figure 6E). Labeling with CBM28 was weaker, however, compared to CBM3a. In the cell walls of the non-inclined plants, the weak labeling of amorphous cellulose was smeared and attributed to the primary cell wall and partially to the thickened layers. In the fibers of the gravistimulated plants (pulling side), amorphous cellulose was presented as the thinner layer. Crystalline cellulose was well detected by CBM3a in both the control and gravistimulated plants in the cell walls of the gravistimulated plants. The intensity of labeling with CBM3a was higher compared to that of the control plants, which was confirmed by the corrected total cell fluorescence (CTCF) measurements (Figure 6E). This observation was less pronounced in the inner layer of the cell wall (i.e., in the newly deposited G-layer (Gn) that had not yet transformed into the mature G-layer (Figure 6, indicated by arrows)). The presence of two layers (Gn and G) in the tertiary cell walls of the flax fibers and their transformations were reported earlier [22,29].

#### 2.3.2. Solid State NMR

To analyse the crystalline interiors, para-crystalline cellulose, and accessible and inaccessible fibril surfaces, we used the methods of cross-polarization/magic angle spinning (CP/MAS) ^13^C NMR and single-pulse (SP)/MAS ^13^C NMR. The preparation of the cellulose samples did not entail any strong chemical influences and included washing the phloem fibers with an ethanol, chloroform, and detergent buffer. The majority of the membranes and intracellular content were also removed by washing. Since the analysed flax fibers were taken at an advanced stage of development, they were characterized by their thick tertiary cell wall with a high cellulose content. The obtained NMR spectra were similar to the spectra of the pre-purified cellulose of flax phloem fibers [30], and the signals in the field of phenolic and protein compounds were not revealed.

#### 2.3.3. CP/MAS ^13^C NMR

The CP MAS ^13^C NMR spectra of cellulose samples from the control and gravistimulated (pulling side) flax plants were very similar (Figure 7). Both spectra showed that the analyzed samples mainly contained cellulose I. The ratios I_α_/I_β_ equal to 0.65 and 0.53 (found for the control and pulling sample, respectively) correspond to the values that are usually assigned to non-lignified materials [31]. Based on the methods of spectral fitting proposed for cellulose I by Larson et al. [32] and Wickholm et al. [33], we performed a detailed deconvolution of the C4 signal region as the most informative region for the analysis of crystalline and paracrystalline cellulose as well as fibril surface. As a result of signal splitting, the four signals (89.7 (I_α_), 89.0 (I_α+β_), 88.7 (para-crystalline), and 88.2 (I_β_) ppm) were referred to the crystalline core. Two signals (84.5 and 83.4 ppm) were related to the accessible fibril surface. The signal at 83.9 ppm was assigned to the inaccessible fibril surface and the signal at 82.3 ppm was ascribed to hemicellulose or short-chain cellulose (Figure 7). According to a quantitative analysis of the spectral region of ^13^C, the CP/MAS NMR spectra of the control and the pulling part of the flax stem revealed that the proportion of crystalline cores and fibril surfaces in the analysed samples was comparable to that previously detected for flax plants [30]. Moreover, the crystalline core fraction and index of crystallinity were increased by ~3% for the cellulose of the gravistimulated plants compared to the control ones (Table 1). In addition, the samples of the pulling part of the stem differed from those of the control by presenting an increase in the proportion of β-allomorph, a slight decrease in α-allomorph, a decrease in the proportion of para-crystalline cellulose, and an inaccessible fibril surface. All observed effects ranged within 2–5%.

Based on the described models of the supramolecular structure that consider cellulose as a fibril with a square cross-section and various fibril aggregates [33,34,35], the average width of the fibrils as well as their lateral aggregates were calculated. The lateral fibril aggregate dimension (LFAD) for the control and pulling sample was the same and amounted to about 15 nm. A slight increase in the fibril width was observed for the sample of the cellulose fibers from the pulling side (Table 1). However, this increase was detected only when applying the aggregate model described by Wickholm et al. [33].

#### 2.3.4. SP/MAS ^13^C NMR

Since CP/MAS NMR suppresses signals from relatively mobile cellulose molecules [36], direct polarization (SP/MAS) with different durations between pulses can be used to produce a more detailed analysis [30]. At a time between pulses of 7 s (Figure 8A), the ratio of the integral intensities of signals belonging to C4 of the interior and surface anhydroglucose units changed in favor of increasing the contribution of signals from the fibril surface and less-ordered cellulose (80–87 ppm) compared to the data of CP MAS ^13^C NMR, where crystal-interior cellulose dominated (88–90 ppm, Figure 7).

The proportion of signals in the C4 region relative to other signals in the SP/MAS spectrum of the cellulose from the pulling side of the stem was reduced by two times when compared to the control plants (Figure 8A). The ratio of the fibril surface signals (C4s) to the crystalline cores (C4i) was not reliable in both analysed samples (Figure 8D). This could be due to a decrease in the mobility of cellulose molecules in the fibers of the pulling stem side both in the fibril surface (C4s) and in the region of crystallites (C4i) when compared to the cellulose from the control plants. As a result, the shorter delay time between signals becomes insufficient for the relaxation of all nuclei of the system, especially the nuclei of carbon atoms localized in the well-ordered interior region of cellulose. With an increase in the delays between pulses of up to 50 s, an increase in the C4 signal intensity of the crystalline cores for both the control and pulling samples was observed (Figure 8B). The ratio of the integral intensities of the C4 signals of the fibril surface (C4s) and crystalline cores (C4i) for the control and gravistimulated plants, thus, became comparable (Figure 8D). For the pulling side samples, along with an increase in the intensity of the C4i signal, the intensity of the C4 of the fibril surface (C4s) also substantially increased in comparison with the control (Figure 8A,B).In the SP/MAS NMR experiments, it was shown that the intensity of the C6 signal of the anhydro-glucose units of cellulose is the least dependent on changes in the recycle time and that nuclei C6s in flax cellulose is characterized by a short relaxation time [30]. Consequently, with a decrease in the recycle time (even a significant one), the C6s nuclei have time to relax, and the corresponding signal is not eliminated from the spectrum, even at a recycle time of 0.1 s. In this regard, to quantify the relative changes in the mobility of the cellulose molecules in the crystalline cores and the fibril surfaces for the control and pulling samples at the different times between pulses, we compared the ratio of the integral intensities of the C6 signal (6s, 62 ppm) and C4 signals (4i, interior, 89 ppm, and 4s, surface, and 84 ppm) (Figure 8C). The ratio of the integral intensities of the C6s and C4 signals of the control and the pulling part samples presented a different tendency depending on the delay time between pulses. For a shorter time, the ratio of these signals in the pulling part samples was several times higher than that in the control plants (Figure 8C, 7 s). This effect was expressed to the greatest extent for the ratio of the signals of C6s and C4i, which was achieved mainly due to a decrease in the proportion of C4i with a short time between pulses. When the time between pulses was increased to 50 s, the ratio of these signals changed mainly due to an increase in the signal intensity in the C4 region. The most dramatic changes were noted for the pulling side of the stem. When, for the control plant samples, changes in the signal ratio with an increase in the delay time were observed only for the ratio of C6s/C4i. Then, in pulling side samples, the integral intensity of the C4 of the fibril surface was also significantly increased. As a result, the ratios of C6s/C4 (both C6s/C4i and C6s/C4s) in the pulling side samples sharply decreased and did not exceed the ratios for cellulose from the control plants (Figure 8C).

An increase in the time between pulses also revealed the differences in the C2 and C6 signals of cellulose between the control and gravistimulated plants. Only in the pulling sample was a simultaneous decrease in the signal intensity of C2 (s, i), C5 (i), and C6 (s) was observed, which was confirmed by the spectra (Figure 8A,B) and the ratio of the signal intensities of C2 and C4i (Figure 8D).

## 3. Discussion

By comparing the flax fiber from gravistimulated and non-gravistimulated plants using biochemistry, immunocytochemistry, and NMR approaches, we revealed the following:
(1)The yield of non-cellulosic polysaccharides (TFA-extracted) was increased in flax fibers subjected to gravistimulation (Figure 2A). This increase was associated mainly with an increase in the content of galactose and galacturonic acid (Figure 2B).(2)In fibers of the gravistimulated plants, epitopes of the LM5 antibody specific for β-(1,4)-d-galactan were localized in a wider cell wall layer compared to the control plants (Figure 3B,C), and the epitope of JIM7 specific for high methyl-esterified homogalacturonan was more abundant in the outer cell wall layer in fibers of the gravistimulated plants compared to the control plants (Figure 4N–Q). Epitopes of LM25 specific for galactosylated xyloglucan were revealed in the fiber cell walls of the gravistimulated plants without galactanase pretreatment (Figure 5B,E) while, in the control plants, epitopes of LM25 became visible only after galactanase pretreatment (Figure 5C).(3)Some changes in the cellulose structure in the flax fibers of the gravistimulated plants were revealed. The distribution of the epitopes of CBM28 and CBM3a antibodies, specific for amorphous and crystalline cellulose, respectively, as well as the signal intensity, was different in the fiber cell walls of the gravistimulated and non-gravistimulated plants. The CP MAS ^13^C NMR spectra revealed that the crystalline core fraction and index of crystallinity were increased for the cellulose of the gravistimulated plants. Furthermore, an increase in the proportion of β-allomorph, a slight decrease in α-allomorph, a decrease in the proportion of para-crystalline cellulose, and inaccessible fibril surface were observed in the samples of the pulling part of the stem compared to the control plants (Table 1). The proportion of signals in the C4 region relative to other signals in the SP/MAS spectrum of the cellulose from the pulling side of the stem was reduced by two times compared to the control plants (Figure 8A,C). The largest decreases in intensity were found for the signals of the fibril surface with a short relaxation time.

### 3.1. Gravistimulation Affects the Content and Rearrangement of Non-Cellulosic Polysaccharides in the Fiber Cell Wall

Plant fibers, which are a mechanical element of the tissue, implement their mechanical functions by forming a thickened cellulosic cell wall to support the vertical position of the stem during plant development [19] and likely act as “motor” cells during the restoration of the vertical position after inclination [8,15]. Gravistimulation (inclination of plants) is an extreme mechanical effect on a plant body during the formation of a response to which some processes associated with the rearrangement of the cell wall might be activated. When studying the consequences of gravistimulation on the cell wall, it is necessary to consider that the revealed changes could be a result of both the modification of already deposited cell wall layers and a consequence of a change in the biosynthesis of newly deposited components, which are usually a combination of these two processes. The temporary upregulation in the expression of cellulose synthase genes in the fibers from the pulling side [37] can be considered evidence for the activation of some biosynthetic processes during gravistimulation.

The slight increase in the content of non-cellulose polysaccharides in the cell walls of the fibers during gravistimulation, as revealed by the biochemical analysis (Figure 2A,B), may indicate the activation of their biosynthesis, which is especially true for pectins (RG-I and HG). It was previously reported that rhamnogalacturonan I (RG-I) with galactan side chains plays a special role in tertiary cell wall formation and functioning [38]. RG-I is synthesized in the Golgi apparatus with long galactan chains [22]. Within the cell wall, RG-I is subjected to shortening of the galactan side chain by specific β-galactosidase [38] and is entrapped between the cellulose microfibrils. This modification of polysaccharides corresponds to the gradual transformation of the newly deposited G-layer (Gn) to a G-layer of a higher density [22,29]. In this regard, two cell wall layers with different levels of labeling are usually detected on the cross-sections of flax fibers: (1) the strongly labeled LM5 layer, which is closer to the plasmalemma (the galactan in it is more accessible for binding to the antibody) and (2) a less bright outer layer, where galactan has shortened chains and is less accessible to the LM5 antibody [38]. In the pulling stem sides of the gravistimulated plants (widen fibers), the width of the inner layer labeled by the LM5 antibody was increased when compared to the control plants (Figure 3C), while the total thickness of the cell wall was decreased [8]. This fact can be explained better by a rearrangement in the structure of the G-layers and Gn-layers than by the active synthesis of a new portion of the Gn-layer. However, according to the monosaccharide analysis, the amount of galactose increased in the cell walls of the gravistimulated plants. The accessibility of RG-I may be induced by a rearrangement of the cell wall, which is associated with changes in the orientation of fibrillar structures. In fibers of the pulling stem side, thick fibrillar structures (500–700 nm) became more pronounced and were oriented 15–30° to the cell axis, while, in the control plants, such structures had an orientation nearly parallel to the cell axis [8]. The accessibility of galactan chains to the LM5 antibody could also be facilitated by the action of cell-wall proteins such as expansins, extensins, and fasciclin-like arabinogalactan proteins, which might loosen the cell wall. This process is similar to the loosening during cell wall expansion. The up-regulation of some genes of the listed proteins was also detected in the fibers of the pulling side of the flax stem [15]. The rearrangement of the cell wall during gravistimulation may also be associated with hormonal changes as well (such as ethylene, gibberellin [7]). Assumptions about the role of cell-wall proteins and phytohormones in the rearrangement of the cell wall during gravistimulation must be elucidated further.

In the fiber cell wall of gravistimulated plants, galacturonic acid was increased when compared to the control plants (Figure 2B). This galacturonic acid is likely attributed to another pectin polysaccharide—homogalacturonan (HG), which is the polymer that usually enriches the primary cell wall and middle lamellae and is essential for cell–cell interactions and conjunctions [39]. According to immunolabeling, HG was concentrated mainly in the middle lamellae and primary cell wall, while very few epitopes of HG were detected in the G-layer (Figure 4). Using two types of antibodies that recognise the different levels of methylation of HG, we demonstrated an increase in the labeling of highly-methylated HG content in the cell walls of gravistimulated plants, but there were no significant differences in the labeling of low-methylated HG in both samples (Figure 4). According to the transcriptomic data, up-regulation of the genes encoding pectin methylesterase inhibitor superfamily proteins (PMEIs) was observed in the pulling side of the flax stem [15]. PMEIs regulate pectin methylesterase (PME) activity [40]. PMEs are enzymes that de-esterify pectins, (mainly HGs), and, thereby, affect the pectin’s properties and cell wall rigidity [41,42]. After demethylation, the cross-linking of HG and other polymers occurred, mainly through calcium. As a rule, this cross-linking increased cell wall rigidity [43]. Thus, the increase of GalA content in gravistimulated plants (as shown in the monosaccharide analysis, Figure 2B) may be due to the synthesis of a new portion of highly-methylated HG, whose degree of methylation can be regulated by PMEI. However, the main pool of HG is located in the primary cell wall and lamellae and can also be subjected to rearrangement during gravistimulation, which, thereby, increases the accessibility of labeling highly-methylated HG. The level of HG methylation also affected water retention in the cell wall, which also influenced the cell wall’s biomechanical properties [44].

Among the cell wall enzymes, genes for xyloglucan endotransglucosylase/hydrolases (XET/XTH) were upregulated in fibers upon plant gravistimulation [15]. Previously, XET activity was found between the S-layers and G-layers during tertiary cell wall formation in tension wood [45]. It was hypothesised that XET modifies xyloglucan, which acts to staple the S-layers and G-layers [45,46]. However, until recently, there were controversial data on the presence of xyloglucan in the G-layers of tension wood [47,48,49,50]. Nevertheless, Kim and Daniel [50] provided clear evidence for the presence of xyloglucans in both mature and developing G-layers. The presence of xyloglucan in the G-layer of the cell wall of the flax fibers has not been reported yet. In the current work, we used two monoclonal antibodies, LM15 (XXXG, [26]) and LM25 (XXLG, XLLG, [27]), specific for different xyloglucan fragments. The LM15 labeling was very weak, or even absent, on the cross-sections of the control and gravistimulated plants (data not shown). According to literature, xyloglucan epitopes in plant cell walls can be masked by other polysaccharides such as by pectic homogalacturonan [26]. The pre-treatment of the cross-section with pectin lyase did not affect the results of the labeling with LM15 and LM25 (data not shown). Since the G-layer is enriched in galactans, we treated the cross-sections with galactanase to remove the potential masking polysaccharide. Using the LM25 antibody specific for galactosylated xyloglucan, we detected xyloglucan in the G-layer of the control plants (Figure 5C). The labeling of xyloglucan in the G-layer was detected in the fiber cell walls of the gravistimulated plants, even without pre-treatment (Figure 5B). There was no difference in the content of xylose between the control and gravistimulated plants (Figure 2B), which demonstrates the rearrangement of the cell wall during gravistimulation. This leads to an increase in accessibility for labeling. We assume that the nature of this effect is similar to that which we suggested for galactan labeling. This effect may be provided by changing the fibrillar structure expressed in the increase of the angle between the cell axis and fibrillar structures in gravistimulated plants [8]. The possible involvement of cell wall proteins in the rearrangement of the cell wall during gravistimulation also cannot be excluded. The interaction between xyloglucan and galactan in the tertiary cell wall may also exist, which was demonstrated for Arabidopsis [51]. Thus, xyloglucan that has the ability to interact with cellulose could be one of the participants in implementing motor reactions in the fiber cell wall, despite its minor content in the tertiary cell wall.

### 3.2. Gravistimulation Affects the Fine Structure of Cellulose in the Fiber Cell Wall

The gelatinous cell wall formed in the fibers of the plant crops and tension wood [18] was enriched in cellulose and, for a long time, it was considered to be a cell wall composed of pure cellulose [52], despite the fact that non-cellulosic polysaccharides, pectins, and hemicelluloses comprise 5–15% of the gelatinous cell wall [17,53] (cellulose remains one of the main components of this type of cell wall). Recently, the flax cell wall’s ultrastructure and mechanical properties during the retting step and changes in the fine structure of cellulose were described [54]. It was demonstrated that the cellulose crystallinity index measured by XRD and confirmed by NMR increases its crystallinity by 4% with retting (over 19 days), which is mainly due to the disappearance of an amorphous polymer. We studied the changes in cellulose crystallinity that occur in plants, which could be considered a physiological response but not a result of the technological processing of plant materials that takes place after plant harvesting. The aim of our study was to describe how native cellulose reacts during the development of the graviresponse and what its role is in ensuring the motor function of the fibers.

According to the labeling of cellulose with two types of antibodies, we found that the amorphous cellulose in gravistimulated plants (pulling side) was mainly concentrated in the outer part of the cell wall, while high crystallinity cellulose was well-labeled in the G-layer (Figure 6B,D). However, due to differences in the affinity of CBM3a and CBM28 for their ligands (crystallinity and amorphous cellulose, respectively), based on the staining intensity, it was not possible to determine the amount of cellulose for the corresponding type in the cell walls [55].

The fine structures of cellulose in the fiber cell walls of the gravistimulated plants and non-inclined plants were estimated using NMR analysis. According to the CP/MAS ^13^C NMR experiments, the index of crystallinity varied from 64.0% ± 0.3% in the control plants to 67.2% ± 0.6% in the gravistimulated plants (Table 1). The obtained values are close to the reported crystallinity index for mature flax fibers without any treatment (64.55%) [56]. The fibril width of cellulose in the control and gravistimulated plants was large (6.5 vs. 7.0 nm), which confirms the increased crystallite size in the tertiary cell wall [57]. The higher crystallinity in the gravistimulated plants compared to the control showed a slight but significant difference [54,58], especially when taking into account that the experiments were performed only a short period after inclination. In fact, the parameters found from the spectra for cellulose in the fibers of the control and gravistimulated plants did not differ significantly. However, along with index of crystallinity, some other differences indicated an enhancement of cellulose ordering and a decrease of cellulose mobility in the gravistimulated plants. This included a decrease of I_α_/I_β_ and paracrystalline cellulose, a slight increase in the crystallite size at an identical size to the aggregates, and a decrease in the signal intensity of the crystalline cores, especially the fibril surface under a short relaxation time (at a similar proportion of the fibril surface at a long relaxation time, SP/MAS ^13^C NMR experiments) (Table 1, Figure 8). The decrease in the mobility of less-ordered cellulose could be due to its interaction with other polymers and due to a change in the number of intramolecular and intermolecular hydrogen bonds and/or modifications of their conformation. It is known that intramolecular O2···O6 hydrogen bonds require more space and have multiple geometries with partial occupation compared to O3···O5 hydrogen bonds with well-defined positions of the hydrogen [59]. Therefore, the sharp decline in the mobility of the fibril surface cellulose due to a decrease in the intensities of the signals corresponding to C2 and C6s in the pulling sample (Figure 8) could be due to the “enforcement” of hydrogen bonds via these atoms in the less-ordered cellulose. This could also lead to an increased ordering of cellulose and reduced mobility of its chains. The hypothetical model of the fine cellulose structure in the fiber cell wall of the control and gravistimulated plants is presented in Figure 9.

Notably, we did not observe any changes in the size of the lateral fibril aggregate dimension in both analysed samples. As described earlier, during the graviresponse, the cell wall thickness in the fibers of the inclined stems was lower when compared to that of the fibers at similar stem levels in the control plants. Possibly, the reduction in cell wall thickness may have been caused by reducing the content of water in the G-layer [8]. The active loss of water in the G-layer could lead to an increase in the cellulose well-ordering that we observe using NMR. Importantly, some researchers in the biomechanics field consider the generation of slow movements via the shrinking and swelling of cell walls as a relevant topic [60]. Moreover, the additional rigidity of the cell wall may provide a local distribution close to that of the bottleneck structures that are formed in the fibers of gravistimulated plants [8].

## 4. Materials and Methods

### 4.1. Plant Material

Plants of long-fibred flax (*Linum usitatissimum* L. cultivar Mogilevsky from the collection of the All-Russia Flax Institute in Torzhok) were grown in the open air in boxes with a 50cm soil layer on the experimental site (Kazan, Russia, 55°47′34.2″ N 49°07′14.9″ E) under natural daylight with daily watering. The experiments started at 30 days after sowing, which was when the plants were in a fast growth period of development. The plant height at the beginning of the experiment was 18–20 cm. Most experiments were performed with plants forced to incline by gentle attachment of the lower part of the stem in the region of cotyledons to the soil with a metal staple, which orients the stem in a horizontal position. The selection of plant material for biochemical and immunocytochemical analysis was carried out 48 h after the inclination of the plants (or at a similar elevation angle of about 40°). A portion of the material was also selected for immunocytochemistry 96 h after the start of inclination. All experiments were carried out on a 5-cm part of the stem, located 3 cm above the cotyledons (Figure 1). Samples were collected from three (for microscopy) and 50 (for biochemical and NMR analysis) inclined and control non-inclined plants. For microscopy, the whole 5-cm parts of the stems of the control and gravistimulated plants were fixed. For the biochemical and NMR analysis, these parts from the inclined plants were preliminarily divided longitudinally into the opposite and pulling sides (Figure 1). For anatomical observations, stem samples were fixed in 4% paraformaldehyde in 0.1 M phosphate buffer (pH 7.2) at 4 °C (overnight), dehydrated in an ethanol series to 70%, and stored at 4 °C. Transverse sections were made with a razor blade and stained in a 0.5% aqueous solution of toluidine blue for 1 min. Microphotographs were made using the transmitted light with an AxioCam HRc camera (Zeiss, Jena, Germany).

### 4.2. Preparation of Fiber Cell Wall Samples for Biochemical and Nmr Analysis

The stem sections (5 cm) were fixed in boiling 96% ethanol for 15 min. Then, the pulling part of the stem was isolated from the inclined plants (Figure 1). Peels from these fragments and all fixed stem sections of the control plants served as a source of isolated phloem fibers. They were washed in a mortar (by pressing with a pestle) with 96% ethanol (5 times for 2 min). This was followed by 80% ethanol (5 times for 2 min). To obtain purified cell walls, the membranes and intracellular contents were removed by washing with chloroform:ethanol (1:2, *v*/*v*, 5 times for 2 min), 96% ethanol (2 times for 5 min), and phosphate-buffered saline with tween 20 (PBS-T) (5 times for 2 min). The PBS-T was removed via deionized water (washed 3 times) and the deionized water was removed using 96% ethanol (washed 2 times for 3 min). The samples were dried at room temperature to a constant weight.

### 4.3. Monosaccharide Analysis

Purified fiber cell walls from the control and pulling parts of gravistimulated plants were homogenized in liquid nitrogen and dried. Samples (10 mg) were hydrolyzed with 2 M TFA at 120 °C for 1 h and cooled. Hydrolysate was clarified by centrifugation at 2000× *g* for 10 min, and the resulting supernatant was dried to remove TFA and dissolved in deionized water (100 µL) [54]. A pellet that was not hydrolyzed in TFA was further hydrolyzed in sulfuric acid, neutralized with saturated Ba(OH)_2_ [61], centrifuged to remove BaSO_4_, and dried and dissolved in deionized water in the same amount as the supernatant after TFA hydrolysis.

The monosaccharide composition of the samples was analyzed by high-performance anion-exchange chromatography using a DX-500 system (Dionex, Sunnyvale, CA, USA) equipped with a CarboPac PA-1 column (4 × 250 mm, Thermo, Waltham, MA, USA) and a pulsed amperometric detector ED40 (Waveform A, 500 ms). The separation of neutral monosaccharides was carried out in 16.5 mM NaOH in the isocratic mode. For the separation of uronic acids, a linear gradient was used from 23.6 mM NaOH in 0.1 M NaOAc to 40.6 mM NaOH in 0.3 M NaOAc for 10 min. The column temperature was 30 °C with a flow rate of 1 mL for 1 min. The results were analyzed using the PeakNet 4.30 software, according to the calibrations obtained for the monosaccharide standards treated in advance with 2 M TFA at 120 °C for 1 h. The analysis was performed for two independent biological and two analytical replicates. Results from this analysis are conveyed as the means ± SD in the diagrams.

### 4.4. Solid-State Nuclear Magnetic Resonance

The ^13^C CP/MAS and SP/MAS NMR experiments were performed on a Bruker WB AVANCE 400 MHz spectrometer (Bruker, Rheinstetten, Germany). The resonance frequency for the carbon nuclei was 100.62 MHz while that for the proton nuclei was 400.13 MHz. Purified fiber cell wall samples were tightly packed in a rotor made of zirconium oxide with a diameter of 4 mm. The rotor was spun at 5 kHz at the magic angle (54.7°). To solute the CP matching due to high-speed CP-MAS, a ramped-amplitude cross-polarization sequence (RAMP) of CP was used (0–100). All experiments were carried out at room temperature. The spectral width was 38.4 kHz, the acquisition time was 50 ms, and the time-domain points were 2 k. For proton decoupling, the TPPM-15 sequence was used. The pulse duration was 8.5 μs, and the power level was 1.5 dB. For the SP/MAS experiment, the duration of the 90° pulse in the carbon nuclei was 4 μs at a power level of 4 dB. The number of accumulations was about 8000, the contact time was 1 ms, and the time between accumulations (relaxation delay or recycle delay) was 4 s for CP/MAS and 7 and 50 s for SP/MAS.

All obtained spectra were analyzed using the TopSpin 3.6.1 software. The chemical shifts of ^13^C in the anhydroglucose residues of cellulose were determined using the carbon peak of L-alanine as an external reference (δ 177.8 ppm) with respect to tetramethylsilane (TMS). Deconvolution of the C1 and C4 regions of the CP-MAS spectra was carried out according to the approach used by Larsson et al. [32] and Zuckerstätter et al. [62]. Fitting of the spectra was based on the least-squares method using resonance assignments with Gaussian and Lorentzian line shapes to correctly describe both the ordered and less-ordered cellulose. Fitting of the line widths was performed by the deconvolution procedure in the PeakFit 4.12 software.

Cellulose crystallinity was calculated by the spectral fitting of the C4 region. One of the methods was based on the ratio of the total area of the C4 signals of anhydroglucose in the crystal interiors (sum of the area of I_α_, 89.6 ppm, I_α+β_, 89.0 ppm, para-crystalline, 88.5 ppm, and I_β_, 88.2 ppm) to the summarized area of the C4 signals of anhydro-glucose units in the crystal interiors and fibril surfaces (85.4, 84.5, and 83.5 ppm) [36]. Another applied method defined cellulose crystallinity as the sum of the area of C4 signals for the crystalline cores (I_α_, I_α + β_, I_β_, and para-crystalline cellulose [31]).

The surface-crystalline cellulose lateral dimensions (LD, nm) of the fibrils and their aggregates were calculated using Equation (1).
(1)$$LD=0.57\times \left(\frac{2+\sqrt{4-4q}}{q}\right).$$

To calculate the lateral dimensions of the fibrils, the fraction q was determined as: (a) the summarized fraction of accessible and inaccessible fibril surfaces and overlapping non-crystalline material (signals at 83.4, 83.9, and 84.5 ppm), i.e., the fibril aggregate model [33] and (b) the ratio of the accessible surface fraction (83.4 and 84.5 ppm) to summarize the accessible surfaces and inner crystalline fractions (88.2, 88.7, 89.0, and 89.7), i.e., the fibrillar model [34]. The lateral fibril aggregate dimensions (LFADs) were calculated via q as a fraction of accessible fibril surfaces only [35]. Crystallite surfaces were considered as monomolecular layers of cellulose directly attached to the crystalline core with a thickness of 0.57 nm [62,63]. The number of biological replicates was 2. For the statistical analysis of gravistimulated plants, data for plants with a similar elevation angle (~40°) were compared. Data errors in the table and figures correspond to the standard deviation. The spectra were assigned according to References [30,31,32,62].

### 4.5. Fluorescence Microscopy of the Stem Sections

The stem samples (5 mm) were fixed in a mixture of 3% paraformaldehyde and 0.5% glutaraldehyde in a PEM buffer (50 mM Pipes, 5 mM EGTA, 5 mM MgSO_4_, pH 6.9) overnight at 4 °C. Then, the samples were dehydrated in a graded aqueous ethanol series and acetone, immersed in LR White resin (Medium Grade Acrylic Resin, Ted Pella, Redding, CA, USA), and added in the proportions (*v*/*v*) of 1:4, 2:3, 3:2, and 4:1, with each step involving a 24-h incubation. Then, the samples were embedded in pure LR White resin in Beem capsules and polymerized at 60 °C for 24 h. Semi-thin sections (1 µm thick) were prepared using a glass knife on an LKB 8800 ultramicrotome (LKB Instruments, Stockholm, Sweden) and collected on silane-coated microscope slides.

For immunohistochemical detection, sections were incubated in Na-phosphate buffered saline (PBS), pH 7.4, containing 3% (*w*/*v*) bovine serum albumin (BSA) for 1 h to block non-specific labeling. Then, the sections were incubated for 1 h with primary antibodies LM5, JIM5, and JIM7 diluted 1:10 and LM25 and LM15, diluted 1:5. The anti-rat IgG linked to fluorescein isothiocyanate (488 nm) (FITC, Sigma, St. Louis, MO, USA) was used as a secondary antibody at a 1:100 dilution, and the samples were incubated for 1 h in the dark. Enzymatic treatments prior to LM25 and LM15 immunolabeling included 0.1 M sodium carbonate for 2 h at room temperature and endo-β-1,4-galactanase from *Aspergillus niger* (Megazyme) in 0.1 M sodium acetate pH 5.5 for 2 h at 37 °C.

The immunohistochemical detection of crystalline and amorphous cellulose cellulose-binding modules CBM3a and CBM28 were used, respectively. Sections were pre-incubated in Na-phosphate buffered saline (PBS), pH 7.4, containing 3% (*w/v*) bovine serum albumin (BSA) for 1 h to block non-specific labeling. Three-step labeling was carried out when using the his-tagged CBMs. The sections were successively incubated with CBMs at 10 µg/mL for 1.5 h. Anti-his mouse antibody (H1029, Sigma Aldrich) was diluted 1:1000 for 1 h, and Anti-mouse IgG FITC (488 nm) was diluted 1:100 for 1 h and mounted in a CFM-1 mounting solution (Electron Microscopic Sciences, Hatfield, PA, USA).

All antibodies were diluted in PBS + 0,06% BSA. Control experiments were performed via the omission of the primary antibody. The sections were examined using a laser confocal fluorescence microscope (LSM 510 Meta, Carl Zeiss, Jena, Germany). Immunofluorescence was observed using excitation at 488 nm and emissions at 503–550 nm. The transmitted light channel was used to detect anatomical details. All fluorescence detection settings for a single antibody and magnification were kept at the same level. The number of fluorescence images was 5–7.

The measurement of the cell wall thickness (fluorescent layer) was carried out using ImageJ. For each sample, the cell wall thickness for 10–12 fibers from three plants was measured in triplicate. Significance in the difference between the two groups was confirmed by a Student’s *t*-test.

The measurement of signal intensity was carried out using ImageJ software and the original protocol from QBI, The University of Queensland (Australia) developed for Corrected Total Cell Fluorescence (published in Reference [25]). For the intensity of each sample of the cell wall, fluorescence was measured using photos of the cross-sections (green channel) of fibers from three plants (20 measures of the sample and 10 measures of the background). The significant difference between the control and gravistimulated plants was confirmed by a Student’s *t*-test.

On the cross-sections labeled with LM25 antibodies, the fluorescence intensity was assessed using ImageJ software for the entire fiber bundle (8–11 bundles per sample).

### 4.6. Transmission Electron Microscopy of Stem Sections

We used the same samples for fluorescence immunohistochemistry that we used for electron microscopy. Ultrathin sections were mounted on Formvar-coated nickel grids. For immunolocalization, the sections were as follows: (1) blocked (15 min, room temperature, high humidity chamber) in Tris-buffered saline with 0.05% Tween 20 (TBST) plus 3% (*w*/*v*) BSA, (2) incubated for 1.5 h at room temperature with primary antibodies JIM5 and JIM7 diluted 1:50 in TB with 0.06% BSA, (3) washed three times in 20 mM Tris-buffer (TB), (4) incubated (1.5 h at room temperature) with secondary antibody goat anti-rat (British BioCell International, BBI, Cardiff, UK) coupled to 10-nm colloidal gold (Amersham Pharmacia Biotech, Piscataway, NJ, USA) diluted 1:50 in TB with 0.06% BSA, and (4) washed in TB and H_2_O. Control experiments were performed by omitting the primary antibody. Sections were stained with saturated aqueous uranyl acetate. This step was followed by staining with Reynolds’ lead citrate [64]. Observations and microphotographs were made with a transmission electron microscope JEM-1200 EX (Jeol, Tokyo, Japan) operating at 80 kV. Five microphotographs of three plants were made for each sample.

## 5. Conclusions

Thus, we suggest that the implementation of the graviresponse is associated with the structural rearrangement of the cell walls in the flax fibers located on the pulling side of the stem. This rearrangement leads to an increase in the accessibility of non-cellulosic polysaccharides in the cell wall layers and middle lamellae (rhamnogalacturonan-I, homogalacturonan,, and xyloglucan) and affects the G-layer transformation that occurs during normal plant development. Hypothetically, this effect could be associated with a change in the angle between the axis of the cell and the cellulose fibrillar structures [8]. Such changes in fibrillar structures can lead to two multidirectional effects, which include the appearance of a region of local expansion and the appearance of a region of local compaction. This may affect the accessibility to label non-cellulosic polysaccharides and increase the ordering of cellulose, respectively. In addition, the loss of water in the cell wall during gravistimulation could also have a significant effect on its rearrangement as well as the observed effects.

## Figures and Tables

**Figure 1 ijms-21-05322-f001:**
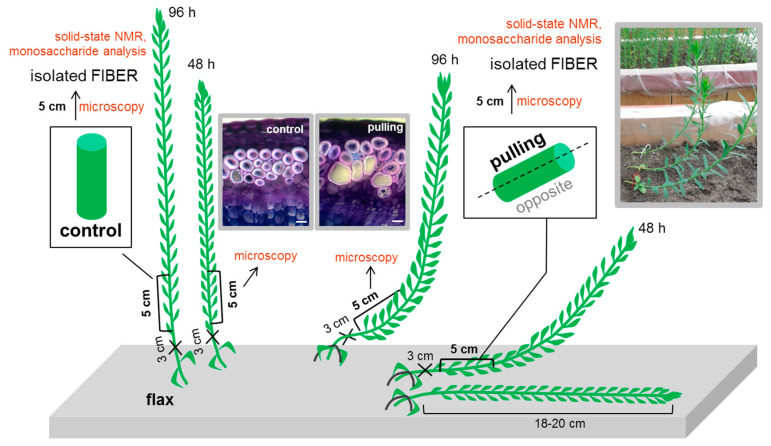
Scheme of the sample collection and analysis. Cross-sections of the control and gravistimulated plants were stained with toluidine blue. Scale bar: 20 μm.

**Figure 2 ijms-21-05322-f002:**
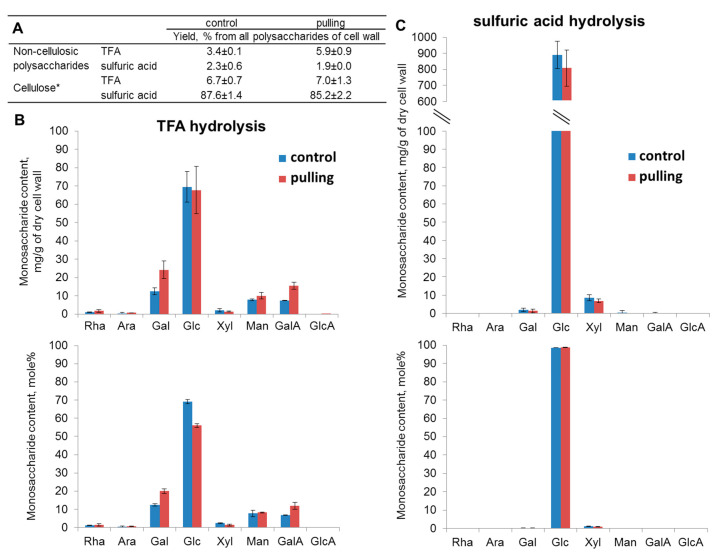
Carbohydrate yield and monosaccharide content in the hydrolysates of the phloem fibers of the control and gravistimulated (48 h after inclination, the pulling part of the stem) flax plants. (**A**)—Yield of non-cellulosic polysaccharides and cellulose from the fiber cell walls of the control and gravistimulated plants after trifluoroacetic acid (TFA) and sulfuric acid hydrolysis. (**B**)—Monosaccharide composition of the TFA hydrolysates of isolated flax fibers. (**C**)—Monosaccharide composition of the residues obtained after the sulfuric acid hydrolysis of pellets pre-treated with TFA. * Cellulose content was calculated from the residual amount of glucose after subtracting its proportion of xyloglucan (Xyl:Glc = 3:4). The xyloglucan-related glucose was then added to the content of non-cellulosic polysaccharides.

**Figure 3 ijms-21-05322-f003:**
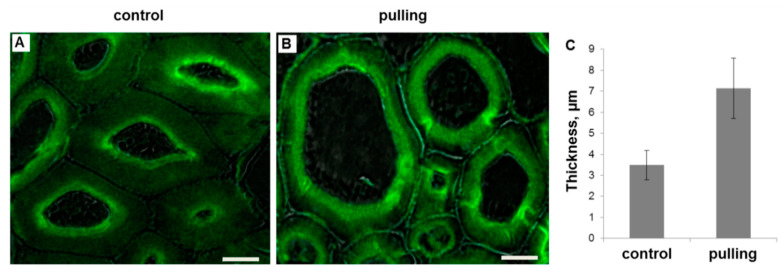
(**A**,**B**)—Immunolabeling of the phloem fibers (cross-sections) of non-inclined (control, **A**) and gravistimulated (96 h after inclination, pulling side of the stem, **B**) flax plants with the LM5 monoclonal antibody specific for β-(1,4)-d-galactan. Scale bar: 10 μm. Fluorescent signal superimposed on the transmitted light image. (**C**)—thickness of the labeled cell wall layer. The measurement of fluorescence was carried out using ImageJ software. For each sample, cell wall thickness of10–12 fibers from three plants was measured in triplicate (*n* ≥ 90 per sample). Error bars: SD. Significance in the difference between the two groups was confirmed by Student’s *t*-test (the *p*-value was <0.00001).

**Figure 4 ijms-21-05322-f004:**
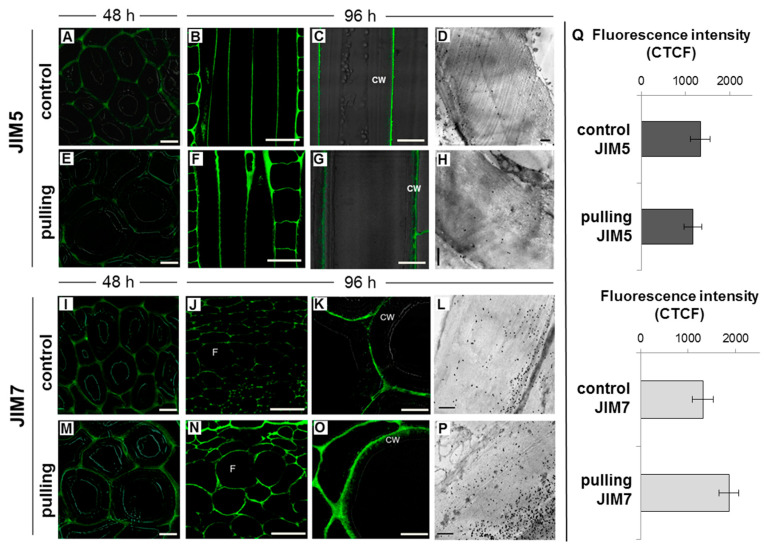
Immunolabeling of low (JIM5, top) and high methyl-esterified (JIM7, bottom) homogalacturonans in the phloem fibers of non-inclined (control) and gravistimulated (pulling side of the stem) flax plants. Labeling of homogalacturonans by JIM5 and JIM7 in the fiber cell wall of the control (**A**,**I**) and inclined (**E**,**M**) plants 48 h after the start of the experiment (confocal microscopy, cross-sections). Labeling of homogalacturonan by JIM5 in the fiber cell wall of the control (**B**–**D**) and inclined (**F**–**H**) plants 96 h after the start of the experiment (**B**,**C**,**F**,**G**—confocal microscopy, longitudinal sections. **D**,**H**— electron microscopy, cross-sections). Labeling of homogalacturonan by JIM7 in the fiber cell wall of the control (**J**–**L**) and inclined (**N**–**P**) plants 96 h after the start of the experiment (cross-sections, **J**, **K**, **N**, **O**—confocal microscopy. **L, P**—electron microscopy). **A**, **C**, **E**, **G**, **I**, **K**, **M**, **O**—fluorescent signal superimposed on the transmitted light image. F—fiber, CW—tertiary cell wall. Scale bars: **B**, **F**, **J**, **N**—50 µm. **A**, **C**, **G**, **I**, **E**, **M**—10 µm. **K**, **O**—5 µm. **D**, **H**, **L**, **P**—1 µm. **Q**—Corrected Total Cell Fluorescence measured for the cell walls of phloem fibers of non-inclined (control) and gravistimulated (pulling side of the stem) flax plants labeled by JIM5 and JIM7. The measurement was carried out using ImageJ based on the original protocol from QBI, The University of Queensland (Australia) [25]. To gauge the intensity of each sample of cell wall fluorescence, the fibers from three plants were analysed (*n* ≥ 60 per sample). Error bars: SD. There were no significant difference between the control and gravistimulated plants according to Student’s *t*-test.

**Figure 5 ijms-21-05322-f005:**
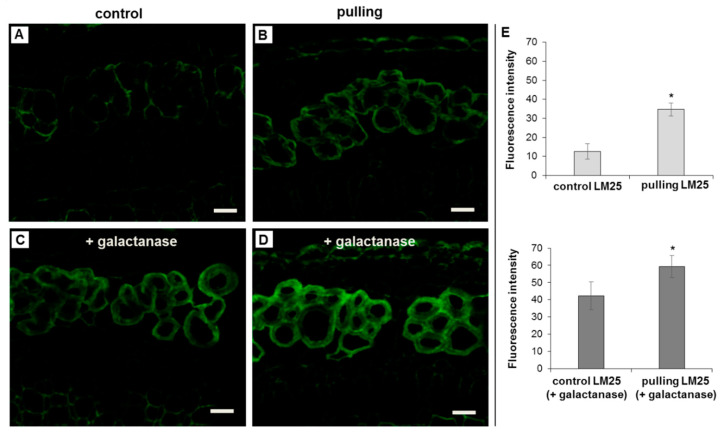
Immunolabeling of the cross-sections of non-inclined (control) and gravistimulated (48 h after inclination, pulling part of the stem) flax plants with the LM25 monoclonal antibody specific for galactosylated xyloglucan. (**A**,**B**)—cross-sections of the control and pulling part of the stem without galactanase pretreatment, respectively. (**C**,**D**)—cross-sections of the control and pulling part of the stem after galactanase treatment of the sections, respectively. Scale bar: 20 μm. (**E**)—intensity of fluorescence of the fiber bundles from three plants (*n* ≥ 8 per sample) calculated using ImageJ (mean gray value). Error bars: SD. *—Significance in the difference between the control and gravistimulated plants was confirmed by a Student’s *t*-test (the *p*-value was < 0.00001).

**Figure 6 ijms-21-05322-f006:**
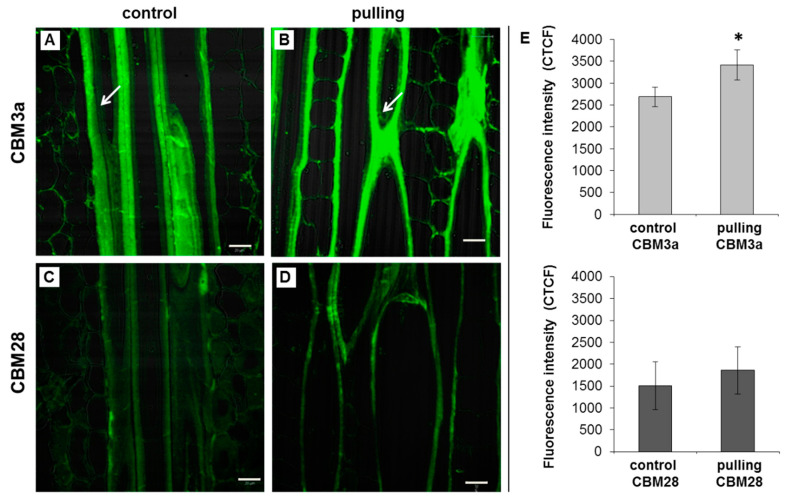
Immunolabeling of high crystalline (CBM3a, **A**,**B**) and amorphous (CBM28, **C**,**D**) cellulose in the phloem fibers of control (non-inclined) and gravistimulated (96 h after inclination, pulling part of the stem) flax plants (longitudinal section). Scale bar: 20 μm. (**E**)—Corrected Total Cell Fluorescence. The measurement was carried out using ImageJ based on the original protocol from QBI, The University of Queensland (Australia) [25]. To gauge the intensity of each sample of cell wall fluorescence, the fibers from three plants were analysed (*n* ≥ 24 per sample). Error bars: SD. *—Significance in the difference between the control and gravistimulated plants (CBM3a) was confirmed by a Student’s *t*-test (the *p*-value was <0.00001).

**Figure 7 ijms-21-05322-f007:**
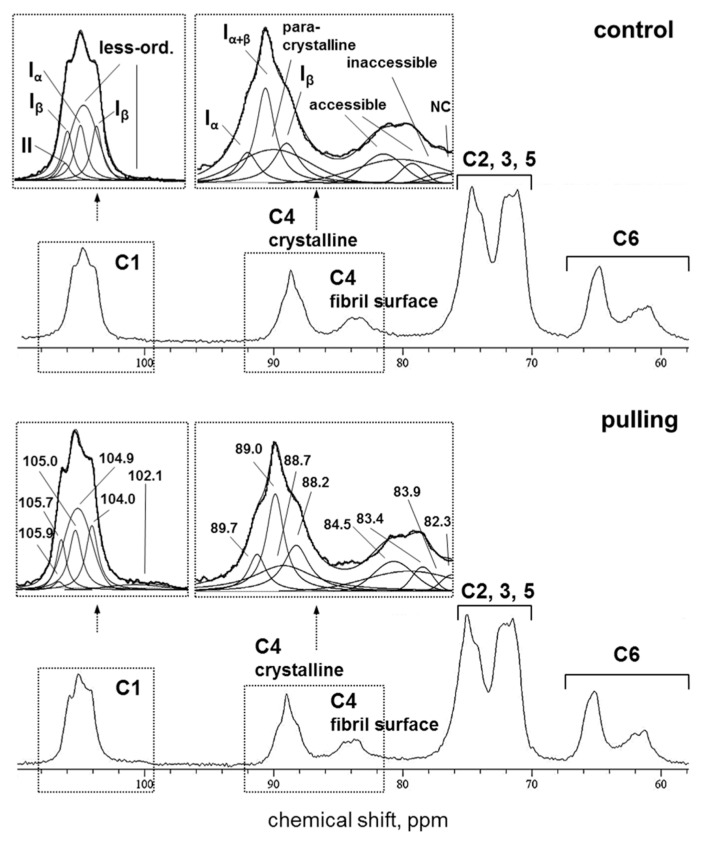
Cross-polarization/magic angle spinning (CP/MAS) ^13^C NMR spectra of cellulose from fibers of the control (top) and gravistimulated flax plants (bottom). Spectral fitting of C1 and C4 regions by the least-squares method using Gaussian and Lorentzian line types is presented in the enlarged areas above the corresponding sections of the spectra. The assignment of signals in the spectrum for cellulose from the control plants corresponds to the chemical shifts presented in the spectrum for the pulling side of the stems of the gravistimulated plants. NC—non-cellulosic polysaccharides or cellulose oligomers.

**Figure 8 ijms-21-05322-f008:**
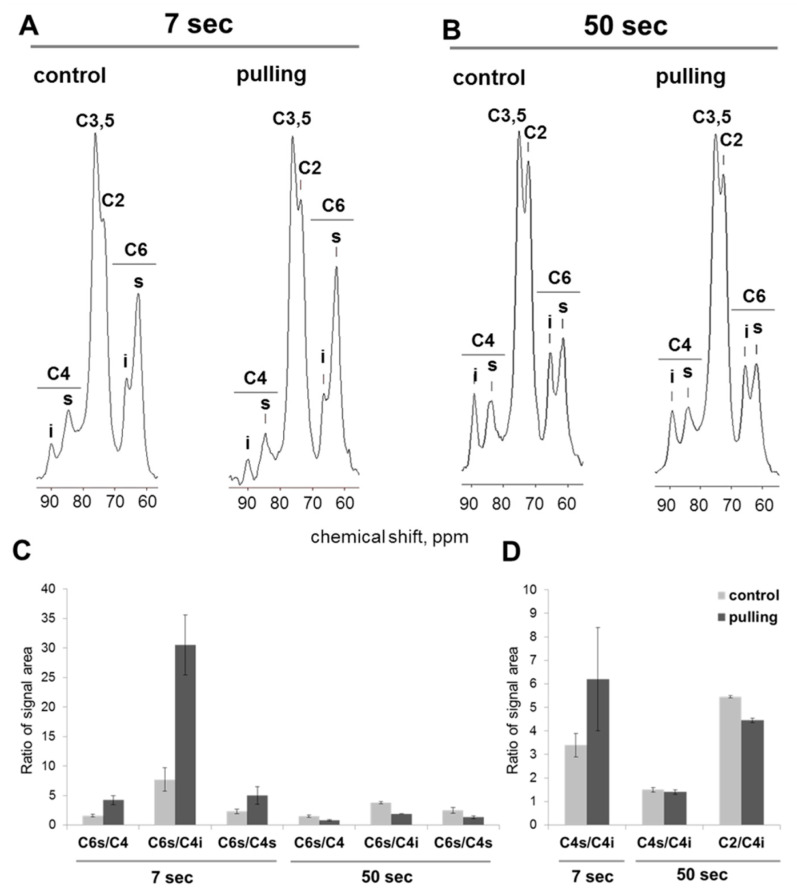
Data of direct polarization (SP/MAS) with different durations between pulses for cellulose from the fibers of the control and gravistimulated flax plants. (**A**,**B**)—SP MAS ^13^C NMR spectra, 7 and 50 s between pulses, respectively. (**C**)—the ratio of the C6s signal area (62 ppm) to the area of the C4 signals (4i—interior, 89 ppm, and 4s-surface, 84 ppm), calculated from the spectra obtained at 7 and 50 s between pulses. (**D**)—the ratio of the fibril surface (s) and crystalline cores (i) calculated from the signal areas of the C4 regions of the spectra at 7 and 50 s and the ratio of the integral intensities of the C2 (i, s), C5 (i) (72 ppm), and C4i signals at 50 s between pulses.

**Figure 9 ijms-21-05322-f009:**
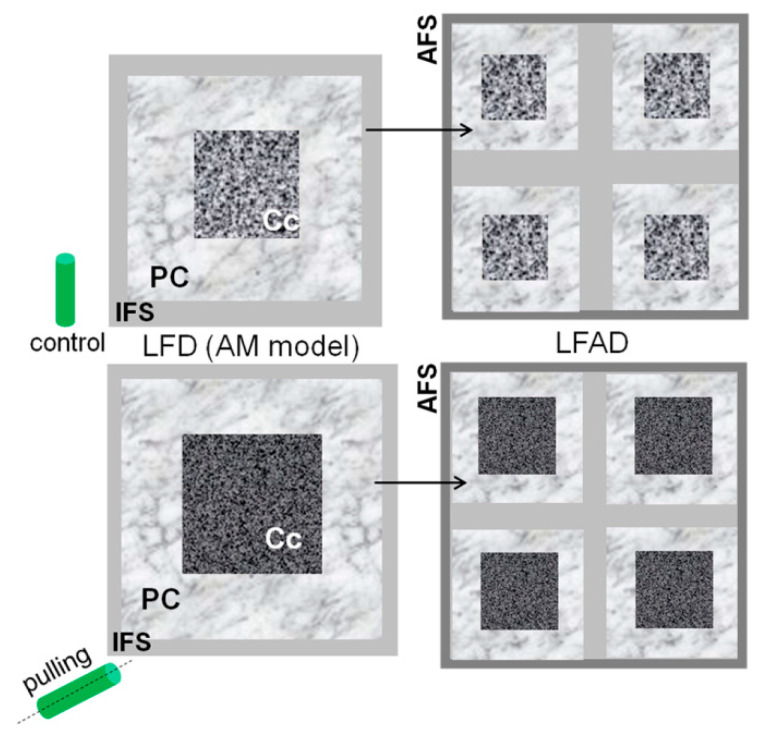
Schematic representation of the flax cellulose elementary fibril structure in the control (top) and gravistimulated (bottom) plants based on the ^13^C CP/MAS NMR experiments. Cc: crystalline cores of cellulose. PC: paracrystalline cellulose. IFS: inaccessible fibril surface. AFS: accessible fibril surface. LFD: lateral fibril dimension. LFAD: lateral fibril aggregate dimension. Gravistimulation of the flax plants (pulling part) was accompanied by an increase in the area of the crystalline cores (Cc) and the fibril width of the cellulose (squares on the left side of the scheme) while retaining the lateral fibril aggregate size (LFAD) and decreasing the proportion of the inaccessible fibril surface (IFS). An increase in the size of the crystalline core at similar aggregate sizes for inclined plants was accompanied by a decrease in the proportion of paracrystalline areas (PC) in their cellulose structures.

**Table 1 ijms-21-05322-t001:** Quantitative analysis of the cross-polarization/magic angle spinning (^13^C CP/MAS) NMR spectra of cellulose samples from fibers of the control and gravistimulated (pulling part of the stem) flax plants.

Experimental Parameters Based on Signal Assignment					
**Sample**	**Crystalline Cores (S_i_), % ***	**I_α_/I_β_**	**Para-Crystalline (PC), % ***	**Accessible Fibril** **Surfaces (S_fs_), % ***	**Inaccessible Fibril Surfaces/Non-Crystalline (S_fs_), % ***
Control	64.0 ± 0.3	0.65 ± 0.00	23.4 ± 0.4	14.2 ± 0.6	17.6 ± 0.2
Pulling	67.2 ± 0.6	0.53 ± 0.02	20.9 ± 0.6	14.4 ± 0.5	15.5 ± 0.0
**Computed Parameters**					
**Sample**	**Fibril Width, nm**		**Lateral Fibril Aggregate Width (LFAD), ^3^ nm**	**Index of Crystallinity**	
				**I_α_ + I_α + β_ + I_β_ + PC**	**S_i_/(S_i_ + S_fs_) * 100%**
Control	6.5 ± 0.1 ^1^	4.5 ± 0.2 ^2^	15.5 ± 0.7	64.0 ± 0.3	66.8 ± 0.4
Pulling	7.0 ± 0.1 ^1^	4.1 ± 0.1 ^2^	15.3 ± 0.5	67.2 ± 0.6	69.2 ± 0.6

* proportion of signal from the area of all signals of the C4 region, ^1^ according to the aggregate model [33], ^2^ according to the fibril model [34], ^3^ according to [35].

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
