# Peer review of "Rearrangement of the Cellulose-Enriched Cell Wall in Flax Phloem Fibers over the Course of the Gravitropic Reaction"

_ijms, 2020, doi:10.3390/ijms21155322_

Round 1

Reviewer 1 Report

The authors provide a description of the cell wall in flax phloem fibers in gravistimulated and non gravistimulated samples. Although the results are not really ground breaking I believe they are novel and I appreciate the effort from the authors to publish this type of informative results.

- However, my main concern is about the lack of quantification of the microscopy images provided. With the images provided it is impossible to assess the potential variability of the observations. It is not clear how many samples were imaged and how representative the images shown are. In my opinion accurate quantification of the signal along with number of sample and statistical test must be provided in order to support almost any of the claims made in the discussion section 3.1.

In addition it would be much more convincing to also provide the full images of the stem section similar to the type of images previously published by the authors in 10.1007/s00709-016-0985-8, fig 3A and B. In the case of gravistimulated samples that could convincingly show differences between the “pulling” and opposite side. Such clear cut qualitative observation may actually reduce the need for quantification in close up images that I suggest above.

- Concerning the NMR results, while the results are quantitative, I could not find any statistical tests of the significance of the observed differences. I am not an expert in NMR, but this still seems like a major problem here, although this could be easily fixed by running the statistical tests on the existing data.

- Line 427: “ Thus, the implementation of graviresponse requires significant structural rearrangement...”. First, here I do not think that the term “require” is appropriate since you do not demonstrate that without these modifications, the gravitropism response doesn’t happen. This should be replaced by something like “is associated with...”. Second, again, unless I missed this information somewhere, there are no statistical tests performed in this study to test the “significance” of the results, so statistical tests should be performed before using this term. Furthermore, the differences observed seem in general very small which make me doubt whether it would actually be significant if tested. Note that even if the statistical analysis would not reveal any significant differences, I believe that this publication would still be worth publishing in this journal. Actually it is quite surprising and interesting that there are so little differences, while the morphological changes are quite major. That would be worth discussing.

Minor comments:

- I globally find that the discussion and some of the results are rather difficult to read. It is quite technical and thorough which of course is fine but difficult to read for non specialist. For instance I am not an expert in NMR and all this section took me very long to read and to grasp what was the interesting information to extract. In addition there are a number of English language mistakes that make it even harder to read. So I would strongly suggest English editing.

- Line 146-148: “The differences in the distribution of low-methylated HG in control and gravistimulated plants detected by electron microscopy were non significant as well (Fig. 3D, H):”. Unless I missed something, I don’t think you can use the term “non significant” here since the data presented is not quantified and no statistical test has been performed to test the “significance”. Similar problem lines 179, 278, 335 and 399.

- Line 314-316: “In the pulling stem side of gravistimulated plants, the width of the inner layer labeled by the LM5 antibody was increased compared to control plants, while the total thickness of the cell wall was decreased.”. Please clarify if this sentence is supported by a reference (e.g. ref 22) or if it is an observation from this publication. If it is the latter, please provide quantification.

I would also like to suggest a few modifications that I believe would make the manuscript easier to read and grasp:

- Start the discussion by a brief summary/overview of the results to ease the later fully detailed description of the results.

- In fig 8 legend briefly explain the model in one or two sentences.

- Separate the last paragraph of the discussion (e.g. 3.3 Conclusion).

- Figure 9 could be displayed at the beginning of the results. It would help a lot in order to understand the design of the study early on to better make sense of the presented results. Maybe around the text line 75-78.

These are only suggestions, I hope they are helpful.

Author Response

Dear Reviewer,

Thank you very much for your review of our article and your highly professional comments. We hope that our revision of the article will satisfy you and we believe that your recommendations have allowed us to significantly improve the manuscript.

We collected all our corrections and improvements in accordance with the comments and wishes of all Reviewers into the Table in the attached file.

Reviewer 2 Report

Firstly I would like to commend the authors on what I found a really interesting piece of work which was very well written, structured and presented. I have a few issues which you can find below, these are extremely minor and will take very little time, no additional wet-lab experimental work is required. Well done!

  • Firstly, I disagree with your use of the cell wall as a ‘motor.’ It is more the tracks or scaffolding on which motors can operate. While the cell wall supplies the mechanical support for the cell which supports adaptation to gravitropism, it cannot do this by itself, the rest of the cell is required to generate and modify the cell wall. Therefore, the cell as a unit is the motor but not just the cell wall. If you inhibited protein secretion and staked plants, no gravitropism would occur as the ‘motor’ effect comes from the combination of cell wall, intracellular wall remodeling and turgor pressure.
  • Secondly, your antibody labelling is fantastic. But I do think some post image analysis could help support your statements. As you state in your methods, you kept all laser power etc the same on the confocal some simply on FiJi you can open your images and calculate fluorescence intensity (ctrl + M). You could also relate this to size of fluorescent elements.
  • For the enzyme work you nicely stated 2 experimental repeats with two technical repeats. I might have missed it but I cant find anywhere the N number of the fluorescent images taken or EM images. If you do not do the analysis as listed above, reporting the N number is very important.
  • I would have expected the work of Brereton et al., to have been mentioned (https://biotechnologyforbiofuels.biomedcentral.com/articles/10.1186/1754-6834-5-83). It was nice work on tension wood in willow.
  • In the discussion, when mentioning your results add the figure number in brackets. Just makes it a lot easier for the reader to refer back.
  • Minor points:
    • Line 11, change ‘at the’ to during
    • Line 374 ‘for’ quite a long time

Author Response

(The authors gave the same response as above.)

Reviewer 3 Report

The article ijms-835404 is an interesting story regarding the rearrangement of cellulose enriched cell wall in phloem in response to gravitropic force. 

Before this study could be reviewed further and considered, there are several technical issues that need clarification. See below

  1. The MS is full of English language mistakes and typos. It will need a moderate language revision. 
  2. The biggest weakness I found in this study is the review of the literature that no report outside the authors' research group has been reviewed and cited. Please remember that the cell-wall is an active topic since the discovery of the cell wall in 1665.
  3. What is the role of the phytohormones in this gravitropic response? It seems that there are several other factors that play an important role in the gravitropic responses in plants.  Considering this, it is very much important to know the soil metal composition. See https://academic.oup.com/pcp/article/61/3/519/5637233.
  4. L300-303. Has this phenomenon i.e. a combination of the two processes i.e. increased biosynthesis of cell wall and rearrangement, already been described in other studies? If yes, cite and discuss. I know the individual processes work in most of the stress responses? This aspect has been totally ignored in the plant recovery process discussed in this study. It was expected to see a bride literature comparison.
  5. Cell wall loosening is completely ignored. Cell wall loosening by the action of expansions is a very much related phenomenon in the recovery process of the plants which experienced gravistimulation. 
  6. The expression of cell wall loosening genes in the parts/sections of the plants selected for the microscopic and biochemical analysis is needed to understand the role of cell wall loosening.
  7. The discussion part 341-346. seems like pure speculation without analyzing the expression of the PMEIs. How did the authors confirm the increased GalA content? Further confirmation is therefore required for the increased content through expression analysis.
  8. Discussion L359-371. needs strong references outside the work of the authors' research group. 
  9. Similar to introduction section, the DISCUSSION section needs to be elaborated with the work on different crop plants. This section of the ms is too weak and is sometimes totally speculation. 
  10. Figures of the original experimental plants should be included if authors want to resubmit in this journal or elsewhere.

Author Response

(The authors gave the same response as above.)

Round 2

Reviewer 1 Report

I find the manuscript greatly improved. I still have a few minor comment that I believe can be quickly corrected:

- I would suggest to also give quantification and statistical tests for the results presented in figure 5.

- To homogenize the representation I feel like figure S1 could be brought into the main text as part of figure 4. That would make it easier to related to image to the quantification and conclusion.

- While I appreciate the effort in clarifying the sample numbers in the figure legends and methods, please provide the actual "n", number of measurements used to plot the graphs and perform the statistical tests in each figure in which measurements are presented. This value should be given in the corresponding figure legend.

- Provide the actual p-values (instead of simply stating that (p≤0.01))

- The * seems to be missing in figure S1

Author Response

Dear Reviewer,

Thank you for continuing to review our article and for your valuable and professional comments and recommendations. We made corrections of the manuscript in accordance with the Reviewer's comments and marked it blue. In the attached table we summarized our responses to Reviewers. We apologize for the delay with our reply.

Reviewer 3 Report

The revised version of the MS ijms-835404 has responded to comments and slightly modified the revised version. The following comments need addressing before the work is approved for the publication.

  1. L42: S2 Layer: Explain any acronym or abbreviation when being used for the first time. If the abbreviation is being used two to three times then it is of no use. Use full term instead.
  2. Previous comment no. 2 about the review of literature and redundancy in citations. The authors proposed the flax as a unique model for the study. The main concern is not the model itself, rather, the limitation of no citations or background information used by authors.

a. L29-33: It is a well-established fact but the reviewer is unable to find any relevant citation/reference.

b. L34-35: Plants (or plant organs) can restore their position in space by involving different mechanisms. For example, roots can have such kind of responses to water availability. https://www.pnas.org/content/111/25/9319.short

c. L34-39: A very detailed review article was published more than a decade ago on the control of gravitropic movements in plants. However, there is no such discussion/citation/report in the current version of the MS. SEE https://doi.org/10.1093/jxb/ern341

d. L34-35: This statement surely should refer to some early work which authors are trying to negate i.e. mainly due to asymmetric growth of the stem in the zone of elongation. ..... It is generally believed that herbaceous plants...... where does this statement come from? 

3. Comment 3 in the first revision. The authors responded well. However, they must introduce one or two sentences in the intro section, so that the story/literature review doesn't seem incomplete or partial. 

4. Comment 4 in the first revision. Authors need to introduce a statement about the limitation or future prospect that .....stress reaction may also contribute to gravitropic responses. 

Author Response

Dear Reviewer,

Thank you for continuing to review our article and for your valuable and professional comments and recommendations. We made corrections of the manuscript in accordance with the Reviewer's comments and marked it blue. In the attached table we summarized our responses to Reviewers. We apologize for the delay with our reply.

Sincerely,

P.Mikshina and co-authors

Round 3

Reviewer 3 Report

Accept

Author Response

Dear Reviewer,

We appreciate your attention to our manuscript and a very helpful discussion. The final file with all the corrections is attached.

Sincerely,
P. Mikshina and co-authors
